# Association of Blood Leukocytes and Hemoglobin with Hospital Mortality in Acute Pulmonary Embolism

**DOI:** 10.3390/jcm12196269

**Published:** 2023-09-28

**Authors:** Slobodan Obradovic, Boris Dzudovic, Bojana Subotic, Sonja Salinger, Jovan Matijasevic, Marija Benic, Tamara Kovacevic, Ana Kovacevic-Kuzmanovic, Irena Mitevska, Vladimir Miloradovic, Ema Jevtic, Aleksandar Neskovic

**Affiliations:** 1Clinic of Cardiology, Military Medical Academy of Belgrade, 11000 Belgrade, Serbia; sloba.d.obradovic@gmail.com (S.O.); bojana.su@gmail.com (B.S.); 2School of Medicine, University of Defense, 11000 Belgrade, Serbia; 3Clinic of Emergency Internal Medicine, Military Medical Academy, 11000 Belgrade, Serbia; 4Clinic of Cardiology, Clinical Center Nis, 18000 Nis, Serbia; sonja.salinger@gmail.com; 5School of Medicine, University of Nis, 18000 Nis, Serbia; 6Institute of Pulmonary Diseases Vojvodina, Novi Sad, 21204 Sremska Kamenica, Serbia; jovanmat99@yahoo.com (J.M.); marija7benic@gmail.com (M.B.); 7School of Medicine, University of Novi Sad, 24000 Subotica, Serbia; 8Clinic of Cardiology, Clinical Center Banja Luka, 78000 Banja Luka, Bosnia and Herzegovina; tamara.kovacevic@medicolaser.info; 9School of Medicine, University of Banja Luka, 78000 Banja Luka, Bosnia and Herzegovina; 10General Hospital Pancevo, 26000 Pancevo, Serbia; anakuzman@gmail.com; 11Intensive Care Unit, University Cardiology Clinic, 1000 Skopje, North Macedonia; peovskai@yahoo.com; 12Clinic of Cardiology, Clinical Center Kragujevac, 34000 Kragujevac, Serbia; vanja.miloradovic@gmail.com (V.M.); ema.jevtic@gmail.com (E.J.); 13School of Medicine, University of Kragujevac, 34000 Kragujevac, Serbia; 14Clinic of Cardiology, University Clinical Center Zemun, 11080 Belgrade, Serbia; neskovic@hotmail.com; 15School of Medicine, University of Belgrade, 11000 Belgrade, Serbia

**Keywords:** pulmonary embolism, mortality, total leukocyte counts, hemoglobin

## Abstract

This study aimed to assess the prognostic significance of total leukocyte count (TLC) and hemoglobin (Hb) levels upon admission for patients with acute pulmonary embolism (PE), considering the European Society of Cardiology (ESC) model for mortality risk. 1622 patients from a regional PE registry were included. Decision tree statistics were employed to evaluate the prognostic value of TLC and Hb, both independently and in conjunction with the ESC model. The results indicated all-cause and PE-related in-hospital mortality rates of 10.7% and 6.5%, respectively. Subgrouping patients based on TLC cut-off values (≤11.2, 11.2–16.84, >16.84 × 10^9^/L) revealed increasing all-cause mortality risks (7.0%, 11.8%, 30.2%). Incorporating Hb levels (≤126 g/L or above) further stratified the lowest risk group into two strata with all-cause mortality rates of 10.1% and 4.7%. Similar trends were observed for PE-related mortality. Notably, TLC improved risk assessment for intermediate–high-risk patients within the ESC model, while Hb levels enhanced mortality risk stratification for lower-risk PE patients in the ESC model for all-cause mortality. In conclusion, TLC and Hb levels upon admission can refine the ESC model’s mortality risk classification for patients with acute PE, providing valuable insights for improved patient management.

## 1. Introduction

The management of acute pulmonary embolism (PE) depends on mortality risk estimation as soon as the diagnosis of acute PE is established and during treatment (1). The best tool for risk stratification is the European Society of Cardiology mortality risk assessment where patients are stratified into four risk strata with different strategies for diagnosis and therapy [1,2]. High-risk patients are hypotensive because of acute PE and need prompt reperfusion therapy to increase their chances of surviving. Patients with right ventricle (RV) dysfunction and elevated blood cardiac troponin (cTn) levels are at the second step on the risk scale, and they need close hemodynamic monitoring for several days and reperfusion therapy in case of deterioration. Patients with intermediate–low-risk have a simplified Pulmonary Embolism Score Index (sPESI) [3] of 1 or more, and may have either RV dysfunction or increased cTn levels in their blood. Low-risk patients have a sPESI of 0, and they are normotensive without RV dysfunction and increased cTn. All patients should be treated with anticoagulation and low-risk patients are candidates for early discharge [1]. However, the fine-tuning of risk assessment is needed. Some prediction signs before the hemodynamic crush should be easy to obtain and applicable to wide clinical settings. The mortality risk in patients with normotensive acute PE is still high, and patients classified as intermediate–high-risk have hospital mortality rates of between 5 and 10%, almost twice as high as patients with ST elevation myocardial infarction (1,2). The routine reperfusion strategy is not recommended for normotensive PE patients because of the risk of excessive bleeding with systemic thrombolysis; however, the data are available only for some thrombolytic protocols and not for other reperfusion options (1). According to current guidelines, reperfusion treatment is recommended only when a patient deteriorates and becomes high-risk; for patients with hypotension and shock the mortality rate is extremely high, and the compensatory mechanisms are already exhausted. So, more sophisticated, simple and reliable methods for risk assessment are needed, especially in the subgroup of patients with right ventricle dysfunction and positive biomarkers for an earlier, pre-shock treatment with some kind of reperfusion therapy.

Total leukocyte blood cell count (TLC) represents an individual, non-specific, inflammatory response to acute illness. It is unknown whether white blood cell increase is only a bystander event associated with a general neuro-inflammatory response, or if it directly contributes to some pathophysiology processes which can impact the outcome [4,5,6]. TLC is associated with higher short-term mortality in acute PE, myocardial infarction, stroke, and sepsis [7,8,9]. The possible use of TLC together with the ESC mortality risk stratification, aiming to create a more precise clinical decision tool, has not been studied before. 

Leukocytes can cause direct damage to the ischemic RV. There are some experimental data on animal models of acute PE, where increased RV pressure induced the expression of various chemokines, and the infiltration of neutrophils into the ischemic myocardial tissue in the early phase, followed by monocytes which differentiate into macrophages with the type 1 phenotype, and in the following weeks into macrophages with the type 2 phenotype [10,11]. These cells caused initial necrosis of the cardiomyocytes caused by myeloperoxidase, esterase and metalloproteinase, and in the subsequent phase fibrosis developed with the impairment of the RV contractile function. A series of autopsies performed on patients dying from massive PE supports the potential role of leukocytes in the damage of RV, where massive infiltrations of neutrophils, lymphocytes and macrophages were found in the extravascular RV tissue, coinciding with myocytolysis [12,13]. A clinical investigation which enrolled more than 14.000 patients with acute PE from the 186 Pennsylvania hospitals showed that total white blood cells had a U-adjusted risk curve for 30-day mortality, while patients with leukopenia (<5.0 × 10^9^/L), and patients with leukocytosis (>12.6 × 10^9^/L), had significantly higher mortality rates compared with patients with leukocyte counts between 7.9 and 9.8 × 10^9^/L [14]. Leukopenia was probably related to comorbidities such as malignancies and infections which per se were associated with adverse outcomes. 

Anemia is very often presented by patients with acute PE at admission, and in the large cohort of more than 14.000 patients it was found in 38.7% of patients and associated with doubled 30-day mortality compared with patients without anemia [15]. Anemia could be associated with higher mortality risk in two different ways: indirectly, as a marker of serious diseases such as chronic heart or renal failure, malignant disease and many chronic inflammatory diseases; and directly, because anemia diminishes the capabilities of the cardiovascular system to deliver oxygen to the organs and tissues, and additionally burdens the compensatory mechanisms and worsens tachycardia and myocardial ischemia. On the other hand, anemia complicates the choice and dosing of the antithrombotic therapy, it represents the hypo-coagulation state, and is a marker of possible ongoing occult bleeding which can become overt and further compromise patients [16]. 

This study aims to evaluate the prognostic value, for short-term all-cause and PE-related death, of TLC and hemoglobin concentrations measured at admission to the hospital for patients with acute PE with respect to the ESC mortality risk stratification.

## 2. Materials and Methods

Data for this investigation were obtained from the Regional PE registry (REPER), a multicenter, international registry of hospitalized patients diagnosed with acute PE using multi-detector computed tomography pulmonary angiography (MDCT-PA). The registry was founded in 2015 and includes data from five university clinics (Military Medical Academy Belgrade, Institute for Pulmonary Diseases Vojvodina, Clinical Centers Zemun, Nis, and Kragujevac), one general hospital (Pancevo) in Serbia, and three university cardiology clinics from Banja Luka (Bosnia and Herzegovina), Podgorica (Montenegro) and Skopje (North Macedonia). 

The main inclusion criteria were: age of 18 and older, symptoms which could be associated with acute PE in the last 2 weeks, hospitalization in the cardiology or pulmonology (only one hospital in the REPER) wards at the time of acute PE diagnosis, and positive findings on admission via MDCT-PA with at least one segmental or three or more subsegmental visualized thrombi (patients with small, or one or two subsegmental contrast defects were not included). The exclusion criterion was hospital admission because of a terminal illness. From a total of 1635 consecutive acute PE patients 1622 had complete blood counts at admission and were enrolled in the study. Patients in this study were enrolled from January 2015 to September 2022. 

All-cause hospital death and PE-related death (where the cause of death was most probably PE) are co-primary outcomes of the study. ICD-10 classification was used for determination of the most probable cause of death. 

The criteria for risk stratification were defined by the 2019 European Society of Cardiology acute PE guidelines [1], where high-risk PE patients have systolic arterial blood pressure of less than 90 mmHg despite the fluid challenge. Intermediate–high-risk PE patients have RV dysfunction, determined via MDCT-PA RV/LV ratio of ≥1 or using echocardiography criteria described in the pulmonary embolism thrombolysis trial (PEITHO study) [17], and also have increased cTn levels in their blood. Intermediate–low-risk patients have a sPESI of ≥1, with or without RV dysfunction or increased cTn blood levels [12]. Low-risk patients are normotensive, have a sPESI = 0, and neither RV dysfunction nor increased cTn levels in their blood. 

Transthoracic echocardiography examination was encouraged on all patients at admission. A total of 1411 patients had RV systolic pressure measured, 1319 had recorded the presence of the McConnell sign, 1236 patients had basal RV diameter at 4-chamber view recorded, and 548 patients had data about tricuspid annular plane excursion at admission recorded. 

Complete blood counts, including TLC, Hb and cTn levels, as an urgent analysis, were measured from the venous blood of all patients at admission to the hospital using standard methods. Brain natriuretic peptide or NT-proBNP and C-reactive protein were measured during the first 24 h after admission, using various standard methods. Upper normal range limits of BNP, NT-proBNP, and cTn levels, were obtained from the laboratories of all hospitals enrolled in the study. 

### Statistics

Description analysis is presented through three subgroups of patients: survivors, all-cause hospital deaths, and PE-related hospital deaths. Basic characteristics of the patients are presented as mean values with standard deviations (age) and as frequencies. Total blood leukocyte counts and Hb are presented as median values with an interquartile range. The Chi-square test is used to establish the significant differences in the distribution of sex and comorbidities between survivors and patients who died during hospitalization. Total leukocyte counts, Hb, and other biomarkers are presented as medians and interquartile ranges, and the differences among these parameters in survivors versus all-cause death and PE-related death subgroups are calculated with the Mann–Whitney test. For the determination of cut-off values for TLC, Hb, and other biomarkers, decision tree statistics were used with the Chi-square automatic interaction detection (CHAID) method for the classification of patients’ risk regarding all-cause and PE-related hospital death. 

Spearman’s correlation test is used between echocardiography parameters and TLC, and Hb levels at admission, with correlation coefficient and *p* values presented in the separate table. 

Univariate and multivariate regression binary analysis was performed using all variables which have significantly different distribution between surviving and deceased patients. For the regression analysis TLC and Hb levels were used as categorical variables with cut-off values calculated from the decision tree statistics. 

Kaplan–Meier analysis and log-rank testing were used to present the hospital survival outcomes in all patients, and in patients who were treated with thrombolysis therapy through the quartiles of TLC and Hb levels at admission. 

Receiver operating characteristics curve analysis for TLC and Hb levels for the prediction of all-cause and PE-related death with areas under the curves, 99% confidence intervals, *p* values and optimal cut-off values and their sensitivity, specificity, negative predictive values and positive predictive values were performed and presented in the Appendix A. *p* less than 0.05 was considered significant for the differences in various parameters between subgroups of patients. 

Statistical analysis was performed using the statistical software IBM SPSS Statistics for Windows, version 20.0 (IBM Corp., Armonk, NY, USA), and ROC curves and statistics were performed using MedCalc version 19.2.6 (MedCalc Software Ltd., Ostend, Belgium).

## 3. Results

The mean age of the study patients was 64 (SD 16) years and 53.3% were women. The basic characteristics of the patients are presented in Table 1 regarding survival status and cause of hospital death. The rates of all-cause hospital death and PE-related hospital death were 10.7% and 6.5%, respectively. Different comorbidities were more often presented in patients who died during hospitalization. Therefore, chronic obstructive pulmonary disease (COPD) (*p* = 0.006), chronic heart failure (CHF) (*p* < 0.001), history of coronary disease (*p* = 0.007), diabetes mellitus (DM) (*p* = 0.001), active malignant disease in the last 6 months (0.004), stroke (*p* < 0.001) and renal failure with a glomerular filtration rate of less than 60 mL/min (*p* < 0.001) were significantly more often present in patients with all-cause hospital death as compared with PE patients who survived. For PE-related hospital death, a significant difference was present for CHF (*p* = 0.04), coronary disease (*p* = 0.01), DM (*p* = 0.011), stroke (*p* < 0.001) and chronic renal failure (*p* < 0.001). There was no difference in the distribution of sex, major surgery, and arterial hypertension between survivors and those with all-cause or PE-related death.

Both TLC and Hb levels were significantly different between survivors and all-cause or PE-related death subgroups (Table 1). TLC was about 2 × 10^9^/L higher in all-cause and PE-related death as compared with the survivors’ subgroup (*p* < 0.001 for both comparisons). On the contrary, Hb levels were slightly, but significantly lower in all-cause death (*p* = 0.001) and PE-related death (*p* = 0.002) subgroups as compared with the survivors.

Using decision tree statistics for the determination of optimal cut-off values of investigated parameters for the prediction of all-cause and PE-related hospital death, our study revealed that both TLC and Hb levels at hospital admission could be used either alone or together with the ESC mortality risk stratification (Figure 1, Figure 2, Figure 3 and Figure 4).

According to TLC, patients can be stratified into 3 subgroups where those who have total leukocyte counts ≤11.2 × 10^9^/L have the lowest all-cause mortality rate of 7.0%, those with TLC between 11.2 and 16.84 × 10^9^ have an 11.8% mortality rate, and the subgroup with TLC higher than 16.84 × 10^9^/L has the highest mortality rate of 30.2% (*p* < 0.001). The first subgroup was further split into two groups using the Hb level at admission. Consequently, the group with admission Hb levels of ≤126 g/L had a mortality rate for all-cause mortality of 10.1%. The mortality rate in the group with Hb levels of >126 g/L was 4.7% (*p* = 0.009). Regarding PE-related mortality, discriminative levels of TLC and Hb levels for the “risk” distribution, were similar to all-cause mortality (Figure 2).

Using decision tree statistics on the ESC mortality risk model, significant discriminative power was achieved between four risk groups such as low-risk, intermediate–low-risk, intermediate–high-risk, and high-risk patients reaching mortality rates of 2.6%, 5.8%, 14.1% and 33.2%, respectively (*p* < 0.001) (Figure 3). Furthermore, patients at the lowest risk were further allocated into two risk subgroups using an Hb cut-off level of 131 g/L, where the mortality rate was 0.7% for patients with Hb >131 g/L, and 4.7% for patients with Hb ≤ 131 g/L (*p* = 0.042). Using a cut-off level of 16.84 × 10^9^/L for TLC we split the intermediate–high-risk subgroup into two risk levels, where patients with total leukocyte TLC levels of >16.84 × 10^9^/L had all-cause mortality of 33.3%, and those with ≤16.84 × 10^9^/L had a mortality rate of 11.2% (*p* < 0.001) (Figure 3). Regarding PE-related mortality in our cohort of patients, the ESC mortality risk model had strong discriminatory power, putting low-risk and intermediate–low-risk patients in the same discriminatory node with a PE-related mortality rate of 1.0%. PE-related mortality rates for intermediate–high and high-risk patients were 8.9% and 28.6%, respectively. In addition, having the TLC cut-off value of 16.5 × 10^9^/L, the intermediate–high-risk group of PE patients were divided into two subgroups where patients with TLC > 16.5 × 10^9^/L had a PE-related mortality rate of 22.6%, and those with TLC ≤ 16.5 × 10^9^/L had a PE mortality rate of 7.0% (*p* = 0.001) (Figure 4). 

Univariate and multivariate binary regression analysis for the prediction of all-cause and PE-related hospital death is presented in Table 2. Tested variables were significantly different between patients who survived and patients who died, either from all-causes or from PE. In the multivariate analysis only variables which have significant independent association with all-cause and PE-related death are present. In the multivariate regression model TLC > 16.5 × 10^9^/L had independent predictive values for all-cause death (OR 3.141, 95% CI 2.062–4.784, *p* < 0.001) and PE-related death (OR 2.738, 95% CI 1.606–4.667, *p* < 0.001) (Table 2). Furthermore, Hb levels > 125 g/L also had independent predictive value for all-cause death (OR 1.775, 95% CI 1.246–2.530, *p* < 0.001) and PE-related death (OR 2.050, 95% CI 1.301–3.320, *p* = 0.002) (Table 2).

Brain natriuretic peptide (BNP) and NT-pro-BNP presented as x at the upper range limit improved the risk stratification in the decision tree statistics of the ESC model (Appendix A). Hence, the PE-related mortality rates of high-risk PE patients were 14.6% and 39.1% for BNP (or NTpro-BNP) levels of ≤8.072 × URL and >8.072 × URL, respectively (*p* = 0.004). 

TLC, Hb level and echocardiographic parameters at admission. The correlation between TLC and Hb levels with some echocardiographic parameters at admission is presented in Table 3. Total leukocyte counts significantly correlate with RV basal diameter, RV systolic pressure and the McConnell sign (positively), and the strongest correlation (negative) was presented with tricuspid annulus plane excursion (Table 3). All of these echocardiographic parameters are well-known parameters of RV strain and they are associated with mortality in acute PE (REF). 

Kaplan–Meier estimates of hospital survival regarding the TLC and Hb level. Kaplan–Meier estimates of hospital survival and log-rank *p* values in respect of quartiles of TLC and Hb levels for the whole cohort of patients, and in cohorts who were treated with thrombolysis (N = 407) are presented in Figure 5A,B and Figure 6A,B, respectively. The fourth quartile of TLC had significantly lower hospital survival rates than the other three quartiles in the whole group and in the subgroup of patients treated with thrombolysis (Figure 5A,B). Additionally, the first quartile of Hb levels had significantly lower hospital survival rates than the other three quartiles. 

Receiver operating characteristics curve analysis was presented in the supplementary analysis (Appendix A). TLC has an area under the curve of 0.649 (*p* < 0.001) for the prediction of all-cause death, and 0.679 (*p* < 0.001) for the prediction of PE-related death. The Hb levels has an area under the curve of 0.580 (*p* = 0.001) for the prediction of all-cause death, and 0.593 (*p* = 0.003) for the prediction of PE-related death. 

## 4. Discussion

In front of the physician who has to treat acute PE patients are two dilemmas: who are the candidates for reperfusion therapy and what kind of reperfusion should be applied, and the second, who are the candidates for early home treatment? Current guidelines do not recommend reperfusion therapy in intermediate–high-risk PE patients before hemodynamic deterioration when the risk of mortality is very high. There is an unmet need for the fine-tuning of the risk stratification which enables reperfusion treatment before the developing overt shock. On the other hand, early discharge to home treatment is challenging in low- and intermediate–low-risk patients. Neither HESTIA rules nor sPESI [18] include anemia as an important factor in the decision about who needs prolonged hospitalization. 

This study showed that two commonly determined, widely used parameters, such as TLC and Hb levels at hospital admission, can be used for the risk stratification of acute PE patients as an addition to the ESC-suggested mortality risk model, and can therefore enhance the treatment decision. Both TLC and Hb admission levels had significant independent association with all-cause and PE-related hospital death in the multivariate analysis, adjusted to ESC mortality risk, creatinine clearance, stroke, and the presence of diabetes mellitus type 2 only for all-cause death. The results showed that using a cut-off level of 16.5 × 10^9^/L of TLC can additionally allocate patients within the intermediate–high-risk PE group into the lower-risk subgroup for TLC value equal to or lower than the cut-off, which had a PE-related mortality rate of 7.0%, and the higher-risk subgroup, with a mortality rate of 22.3% (TLC higher than the cut-off), similar to the mortality rate of high-risk patients in our study. On the contrary, an Hb cut-off level of 131 g/L can be used for the risk stratification of low-risk PE patients where patients with lower Hb levels had an almost five times higher all-cause mortality rate (4.7% vs. 0.7%) than patients with Hb levels above the cut-off value. From the clinical point of view, these results might have two important consequences. The first is that the subgroup of patients with intermediate–high-risk PE with TLC levels higher than 16.5 × 10^9^/L at admission should be considered as higher-risk and treated with reperfusion therapy if there are no contraindications. The second is that patients with low-risk PE and Hb levels, under 131 g/L at admission, should be hospitalized and carefully evaluated because of the increased risk of all-cause mortality.

This study also confirmed significant correlations between TLC and echocardiographic parameters of PE severity [19] at admission which at least potentiate the possible role of leukocytes in the development of RV dysfunction. 

Regarding routine complete blood cell analysis, TLC, neutrophil to lymphocyte ratio, mean platelets volume, and anemia, these are the most studied markers of poor prognosis in patients with PE and other acute vascular disorders [5,6,9,14,20,21,22]. Leukocytosis in acute PE may be caused by local inflammation derived from pulmonary infarction in peripheral PE; however, this kind of leukocytosis has no predictive value for mortality, especially for PE-related mortality [4]. However, in patients with intermediate–high- and high-risk PE, leukocytosis is part of the systemic inflammatory response and catecholamine surge as compensatory mechanisms for severe acute disease [23]. Hence, leukocytosis is a part of the response to acute PE, but has an unknown role in the pathophysiology of the disease. 

There are animal models of acute PE [10,11], and human autopsy studies of fatal PE [12,13] and acute PE. Severe acute stretching and ischemia in acute PE resulted in chemokine synthesis in the RV wall and efflux of neutrophils at first and monocytes and lymphocytes in the second wave in the extracellular space of the RV. Neutrophils produce and secrete various enzymes which could cause cardiomyocyte injury and necrosis with cardiac troponin release which is the blood marker of severe PE and has important prognostic significance. Monocytes go through different phases and phenotypes, at first producing an abundance of chemokines and other pro-inflammatory cytokines which contribute to the inflammatory injury of the RV. Finally, M2 monocytes contribute to fibrosis development and irreversible RV dysfunction in the most severe cases [11].

Leukocytes may have a direct role in the thrombotic process especially in some inflammatory diseases [24,25]. Neutrophils produce neutrophil extracellular traps (NETS) and neutrophil derived microparticles (NMP), which are the core of thrombotic process especially in infections and malignant diseases. NETS and NMP promote platelet adhesion and activation and activation of several proteins in the coagulation cascade. All of this can promote deep venous thrombosis, and PE and thrombosis in heart microcirculation during ischemia, and thus contribute to the adverse outcome of severe acute PE [24,25]. 

Our study has revealed that leukocytosis is especially important for the prognosis of intermediate–high-risk patients where the therapeutic approach is the most difficult and where we need more simple and reliable markers for early prognosis. This study also demonstrated that TLC and Hb levels retained their prognostic value in patients who received thrombolytic therapy. As far as we know, this is the first study which showed this prognostic significance of TLC and Hb levels in severe PE patients who were treated with thrombolysis. 

The largest clinical study regarding the prognostic value of leukocytes in acute PE was by Venetz et al. [14], on 14.228 acute PE patients in 186 Pennsylvania hospitals. The authors divided patients according to modified quartiles of TLC into five groups: <5.0, 5–7.8, 7.9–9.8, 9.9–12.6, and >12.6 × 10^9^/L. The authors recognized that TLC had a prognostic U-curve risk for 30-day mortality, where patients with leukopenia (<5.0 × 10^9^/L) and in the fourth quartile (>12.6 × 10^9^/L) had the worst outcomes, with OR of 1.52 (95%CI 1.14–2.03), and 2.22 (95%CI 1.83–2.69), respectively, compared with the second group that served as a reference with the lowest 30-day mortality. The special advantage of this study, apart from the large number of patients, was the multivariable adjustment of the prediction model, using demographics, comorbidities, hemodynamic and other laboratory parameters, and where TLC remained the independent prognostic marker for early mortality. In our study we also used the multiple regression model to adjust ORs of TLC and Hb optimal cut-off levels gained through the decision tree statistics, and showed that both parameters independently, and adjusted to renal function and the ESC mortality risk model, are the most important confounding factors, predicting all-cause and PE-related hospital mortality. The advantage of our work is that we used the ESC mortality risk model for the adjustment of risk prediction for both all-cause, and PE-related mortality regarding TLC and Hb levels, because the ESC mortality risk model establishes a risk stratification score in acute PE recommended by the ESC guidelines, and it is used as a central treatment strategy for the management of these patients. 

Improvement of the prediction accuracy of some verified prediction models such as the simplified pulmonary embolism severity score (sPESI) by the addition of hematological parameters is a potentially clinically useful concept. In the study by Slajus et al. [22], adding red cell distribution width, hematocrit and neutrophil to lymphocyte ratio significantly improved the accuracy of the simplified PESI score for death prediction in 228 acute PE patients without malignant and chronic inflammatory diseases, which can confound the results. However, an sPESI score is not recommended for the initial stratification of acute PE patients without using the ESC risk model based on hemodynamic status, RV function and biomarkers. In this way, our results are more clinically applicable by re-stratifying patients from intermediate–high-risk to even higher risk of dying, according to leukocytosis levels higher than 16.5 × 10^9^/L, with PE-related mortality of 23%, which is very close to high-risk PE patients, and implies the possible benefits of reperfusion therapy in those patients without waiting for further hemodynamic compromise. 

Moreover, lower Hb levels in acute PE are a consequence of various comorbidities, with or without bleeding. Several reports suggest that anemia and particularly persistent anemia [26] are important risk factors for poor outcomes in acute PE. In our study, we have shown that anemia may be considered as an additional risk for dying in patients with otherwise low-risk PE. 

We tested three other biomarkers with decision tree statistics and found that neither CRP nor cTn add additional predictive value to the ESC model. However, a BNP and NT-proBNP increase in relation to the laboratory determined upper normal limit range has significant value for the prediction of PE-related mortality in high-risk PE patients. However, cTn is already incorporated in the ESC mortality risk model. Several biomarkers have strong predictive value for mortality risk in normotensive PE patients; however, their prognostic value in high-risk patients is not defined [27]. The possible value of this finding might be important since 40% of high-risk patients had BNP or NT-proBNP values less than eight times above the ULR and the mortality rate was very similar to intermediate–high-risk PE patients. 

## 5. Limitations of the Study

This study enrolled only hospitalized patients with acute PE, which means that a small portion of low-risk patients who were treated at home were not included. Additionally, the majority of hospitals enrolled in the REPER include only patients admitted to cardiology wards, and there is a trend toward admitting more severe patients to the cardiology wards, which means that a bias toward enrollment of more severe patients existed in our study. In spite of the fact that physicians made every effort to treat patients according to current guidelines because of the local specificities, we can only approximately generalize these results, although the mortality of the ESC mortality risk subgroups is similar to other investigations and registries [2,28]. 

We had no correction for TLC regarding the volume status, because some patients might have had leukocytosis due to dehydration. The association between the TLC and other markers of hypercoagulability like fibrinogen and coagulation factors were not studied [29].

In this investigation we did not use repeated findings of TLC and Hb and thus we did not investigate the role of the dynamic changes of TLC and Hb in acute PE outcomes.

Some patients were treated with immunomodulatory drugs and chemotherapy, however, which can influence both the TLC and Hb levels at admission, but we did not exclude these patients and did not analyze the potential impact of this on our results. 

A relatively low proportion of patients (about 10%) in the intermediate–high PE group had TLC levels above the threshold of 16 × 10^9^/L. The classification of PE-related death is arbitrary, and, in some patients, a significant uncertainty exists regarding the real cause of death. We also did not use BNP/NT-proBNP levels for the multivariate models since there were a high number of missing values and different assays used for this marker in our study.

## 6. Conclusions

TLC and Hb levels in acute PE on admission to hospital have a significant value for the prediction of hospital death in addition to the ESC mortality risk model. Those parameters could be used in clinical practice during the decision-making process for the treatment of low-risk and intermediate–high-risk PE patients.

Future directions. TLC and Hb levels should be tested as complementary, simple, repeatable risk markers, adding them to the ESC mortality risk model, sPESI or HESTIA rule, for the timely management of severe PE patients with reperfusion therapy, and for making the decision about early safe discharge in low-risk PE patients.

## Figures and Tables

**Figure 1 jcm-12-06269-f001:**
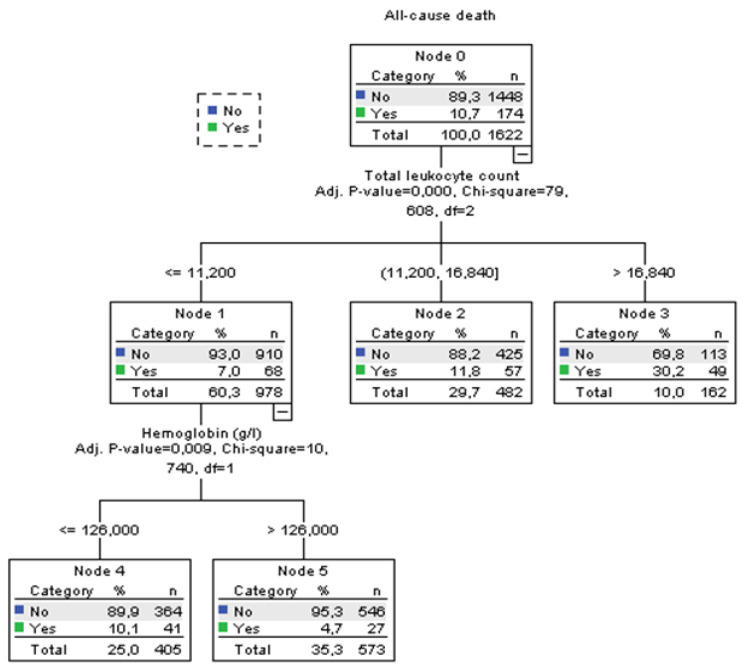
Decision tree for all-cause hospital death in acute PE using blood total leukocyte count and Hb concentration at admission.

**Figure 2 jcm-12-06269-f002:**
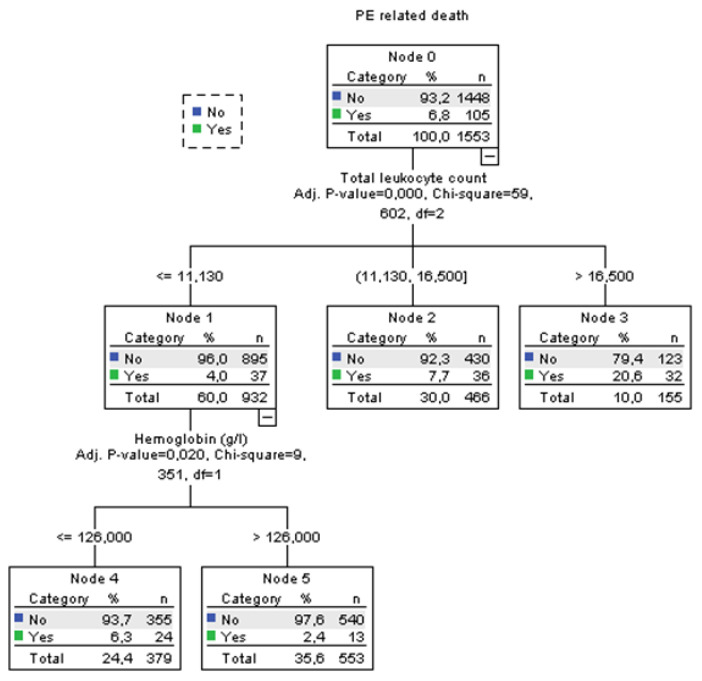
Decision tree for PE-related hospital death in acute PE using blood total leukocyte count and Hb concentration at admission.

**Figure 3 jcm-12-06269-f003:**
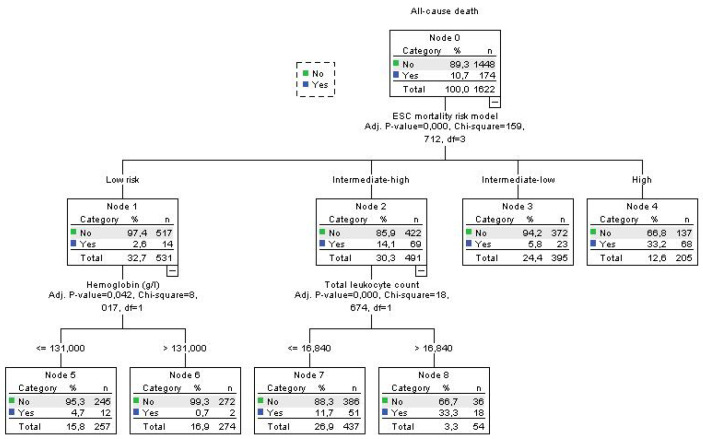
Decision tree for all-cause death in acute PE using blood total leukocyte count and Hb concentration and ESC mortality risk stratification at admission.

**Figure 4 jcm-12-06269-f004:**
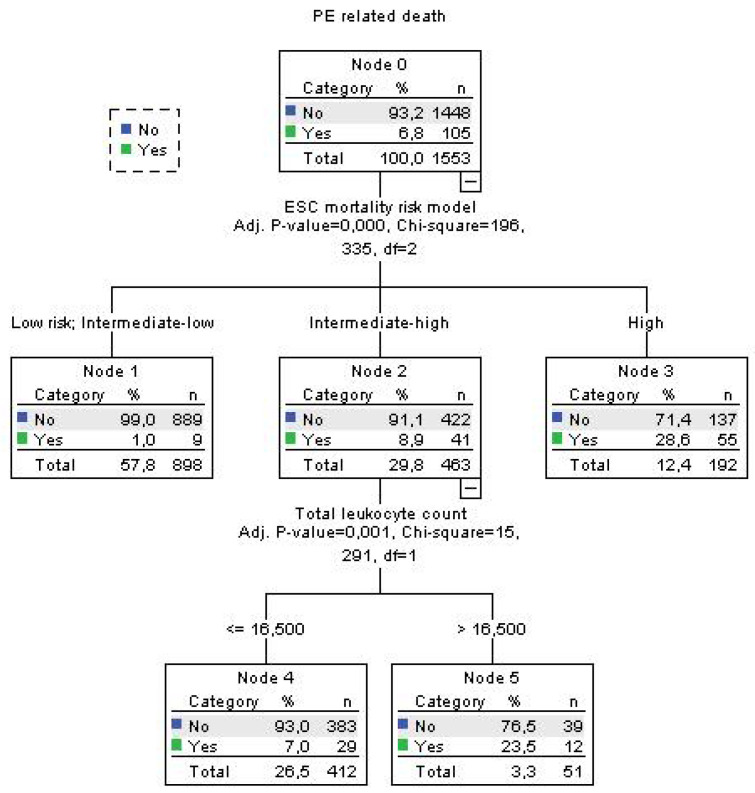
Decision tree for PE-related death in acute PE using blood total leukocyte count and Hb concentration and ESC mortality risk stratification at admission.

**Figure 5 jcm-12-06269-f005:**
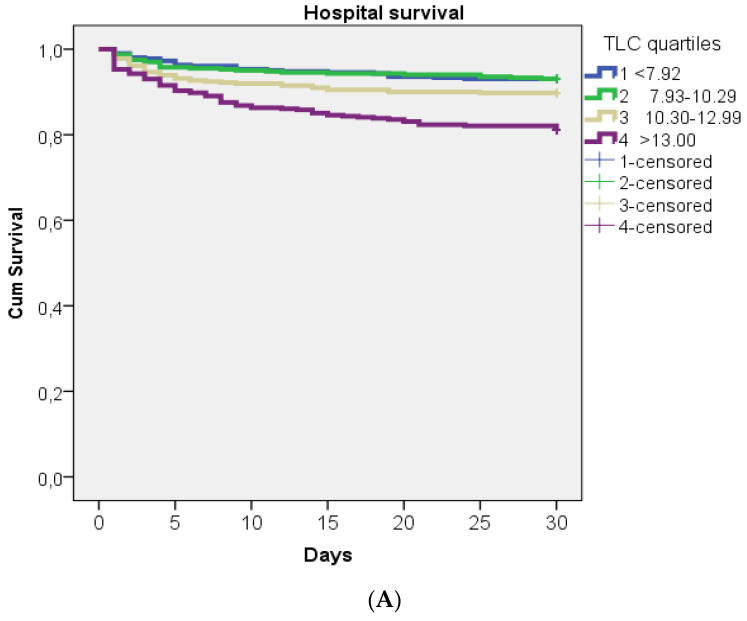
(**A**,**B**). Kaplan–Meier estimates of hospital survival with respect to the TLC quartiles (×10^9^/L) in the whole group of patients (**A**) and in subgroups who were treated with thrombolysis (**B**). For both KM, log-rank *p* is <0.001.

**Figure 6 jcm-12-06269-f006:**
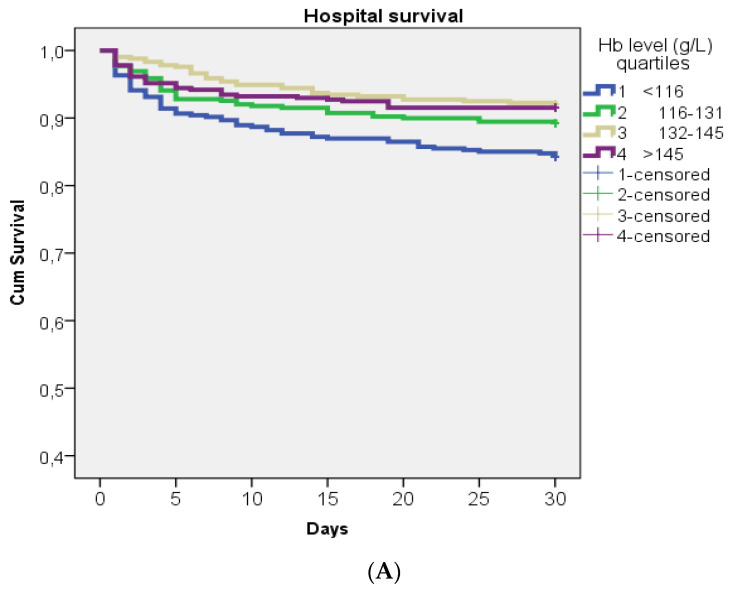
(**A**,**B**). Kaplan–Meier estimates of hospital survival with respect to the Hb level quartiles in the whole group of patients (**A**) and in subgroups who were treated with thrombolysis (**B**). For the whole group the log-rank *p* value is 0.001, and for the Kaplan–Meier analysis of survival in patients treated with thrombolysis *p* = 0.042.

**Table 1 jcm-12-06269-t001:** Basic characteristics of patients at admission to hospital.

	Survivors N = 1448	All-Cause Death N = 174	*p*	PE-Related Death N = 105	*p*
Age, y mean ± SD	63 ± 16	70 ± 14	<0.001	69 ± 15	<0.001
Female—n, %	767, 53.0	94, 55.7	0.520	58, 55.2	0.686
COPD—n, %	142, 9.8	30, 17.2	0.006	13, 12.4	0.398
CHF—n, %	195, 13.5	45, 25.9	<0.001	22, 21.0	0.040
Malignant dis.—n, %	181, 12.5	36, 20.7	0.002	18, 17.1	0.096
Surgery—n, %	224, 15.5	25, 14.4	0.824	14, 13.3	0.674
Diabetes—n, %	274, 18.9	53, 30.5	0.001	31, 29.5	0.011
Coronary dis.—n, %	157, 10.9	31, 18.5	0.007	20, 19.8	0.010
Stroke—n, %	89, 6.2	27, 15.6	<0.001	17, 16.2	<0.001
GRF < 60 mL/min—n, %	474, 33.0	115, 66.5	<0.001	76, 72.4	<0.001
sPESI > 0—n, %	970, 67.0	159, 91.4	<0.001	96, 91.4	<0.001
ESC mortality risk			<0.001		<0.001
Low	517, 35.7	14, 8.0		5, 4.8	
Intermediate-low	372, 25.7	23, 13.2		4, 3.8	
Intermediate-high	422, 29.1	69, 39.7		41, 39.0	
High	137, 9.5	68, 39.1		55, 52.4	
TLC × 10^9^/L median (25th–75th)	10.0 (7.9–12.7)	12.1 (9.3–17.6)	<0.001	12.4 (9.8–17.6)	<0.001
Hb g/L median (25th–75th)	133 (118–145)	125 (110–141)	0.001	123 (107–140)	0.002
BNP or NT-proBNP (×UNRL)	1.86 (0.58–5.85)	6.8 (2.56–17.02)	<0.001	7.33 (3.52–18.80)	<0.001
cTn (×UNRL)	2.0 (0.32–9.06)	4.87 (1.57–16.16)	<0.001	5.06 (2.0–18.3)	<0.001
CRP mg/L (25th–75th)	41.6 (16.1–89.85)	80.50 (44.30–163.6)	<0.001	78.05 (45.85–164.6)	<0.001

COPD—chronic obstructive pulmonary disease, CHF—chronic heart failure, GRF—glomerular filtration rate, TLC—total leukocyte count, Hb—hemoglobin, BNP and NT-proBNP—brain natriuretic peptide, and N terminal brain natriuretic peptide, UNRL—upper normal range limit, cTn—cardiac troponin, CRP—C-reactive protein.

**Table 2 jcm-12-06269-t002:** Univariate and multivariate regression analysis for the prediction of all-cause and PE-related hospital death, using total leukocyte count as the categorical variable with cut-off at <16.5 × 10^9^/L and Hb blood concentration of <125 g/dL. In the multivariate analysis adjustment was performed with variables which have significantly different distribution between survivors and deceased patients. In the multivariate-adjusted analysis, only variables with independent significant association with death are presented.

All-Cause Death	Univariate Analysis OR (95% CI, *p*)	Multivariate Analysis OR (95% CI, *p*)
Age	1.033 (1.021–1.046, <0.001)	-
Chronic obstructive pulmonary disease	1.916 (1.247–2.945, 0.003)	-
Chronic heart failure	2.242 (1.546–3.250, <0.001)	-
Diabetes mellitus type 2	1.877 (1.324–2.660, <0.001)	1.537 (1.039–2.274, 0.031)
Malignant disease	1.903 (1.281–2.826, 0.001)	-
Coronary disease	1.858 (1.216–2.838, 0.004)	-
Stroke	2.822 (1.776–4.484, <0.001)	2.110 (1.237–3.599, 0.006)
Chronic renal failure (CrCl < 60 mL/min)	4.032 (2.877–5.633, <0.001)	2.549 (1.773–3.665, <0.001)
ESC mortality risk model	compare to low risk	compare to low risk
Intermediate–low	2.283 (1.159–4.496, 0.017)	2.441 (1.206–4.945, 0.013)
Intermediate–high	6.038 (3.351–10.879, <0.001)	5.506 (2.963–10.229, 0.001)
high	18.330 (10.007–33.575, <0.001)	13.760 (7.239–26.156, <0.001)
Total leukocyte count (cut-off 16.5 × 10^9^/L)	4.591 (3.160–6.670, <0.001)	3.141 (2.062–4.784, <0.001)
Hb (cut-off 125 g/L) ^1^	1.816 (1.323–2.490, <0.001)	1.775 (1.246–2.530, <0.001)
PE-related death		
Age	1.030 (1.014–1.045, <0.001)	-
Chronic heart failure	1.703 (1.040–2.790, 0.034)	-
Diabetes mellitus type 2	1.795 (1.157–2.785, 0.009)	-
Coronary disease	2.027 (1.209–3.398, 0.007)	-
Stroke	2.948 (1.681–5.170, <0.001)	2.091 (1.083–4.035, 0.028)
Chronic renal failure (CrCl < 60 mL/min)	5.330 (3.427–8.290, <0.001)	3.138 (1.940–5.076, <0.001)
ESC mortality risk model	compare to low risk	compare to low risk
Intermediate–low	1.112 (0.297–4.168, 0.875)	1.112 (0.297–4.239, 0.865)
Intermediate–high	10.046 (3.935–25.648, <0.001)	8.341 (3.233–21.518, <0.001)
high	41.511 (16.302–105.70, <0.001)	29.497 (11.414–76.230, <0.001)
Total leukocyte count (cut-off 16.5 × 10^9^/L)	4.722 (2.997–7.441, <0.001)	2.738 (1.606–4.667, <0.001)
Hb (cut-off 125 g/L) ^1^	2.108 (1.412–3.1.47, <0.001)	2.050 (1.301–3.320, 0.002)

C-reactive protein level and cardiac troponin level did not influence the risk assessment adding to the ESC mortality risk model for either all-cause or PE-related hospital death. Brain natriuretic peptide of NT-proBNP was measured on 980 out of 1662 patients and after that, this variable was not included in the multivariable and ROC curve analysis. ^1^ OR was presented as risk when comparing the subgroup with anemia (Hb < 125 g/L) to the subgroup with normal hemoglobin level.

**Table 3 jcm-12-06269-t003:** Spearman’s correlation between TLC, Hb and echocardiographic parameters at admission.

	Blood Parameters	RV Systolic Pressure	Tricuspid Annulus Plane Excursion	McConnell Sign
TLC				
Correlation coefficient	0.077	0.095	−1.99	0.134
*p*	0.007	<0.001	<0.001	<0.001
N	1236	1411	548	1319
Hb				
Correlation coefficient	0.104	0.052	0.051	0.019
*p*	<0.001	0.052	0.237	0.487
N	1236	1411	548	1319

## Data Availability

Data will be available on reasonable request.

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
