# Peer review of "Association of Blood Leukocytes and Hemoglobin with Hospital Mortality in Acute Pulmonary Embolism"

_jcm, 2023, doi:10.3390/jcm12196269_

Round 1

Reviewer 1 Report

Acute pulmonary embolism (PE) is associated with difficulties in diagnosis, treatment and observation. The clinical course is very variable: from an asymptomatic course to a massive embolism with hemodynamic instability, right ventricular failure and death). Annual incidence rates range from 0.2 to 0.8/1000, and mortality ranges from 17 to 48% within 30 days and from 19 to 30% within 6 to 12 months after an episode of acute PE. All this justifies the need to organize and conduct research aimed at improving early diagnosis and predicting the outcomes of the disease, as well as reducing morbidity and mortality from this pathology, justified from a scientific point of view and in demand from a practical point of view.

As a result of this study, based on the analysis of data from the Regional PE Registry (REPER) and the international register of hospitalized patients diagnosed with acute PE using multi-detector computed tomographic lung angiography (MDCT-PA), the authors examined the predictive value of the total number of leukocytes and hemoglobin levels at admission of patients with acute pulmonary embolism (PE) based on the European Society of Cardiology (ESC) mortality risk model.

Strengths of the study: sufficient sample of patients; well-planned study design; the use of adequate statistical methods, the availability of introducing the definition of the studied parameters and, in this regard, the high potential for practical use of the results, the presence of open questions that reasonably require further research in order to improve the risk stratification of adverse cardiovascular events associated with acute PE, for example, to clarify the role molecular biomarkers.

Weaknesses of the presentation of the manuscript: the lack of p-value of the parameters in Table 1. Moreover, the authors should add the data on the baseline levels of echocardiography and their relationship with the studied parameters.

Author Response

Acute pulmonary embolism (PE) is associated with difficulties in diagnosis, treatment and observation. The clinical course is very variable: from an asymptomatic course to a massive embolism with hemodynamic instability, right ventricular failure and death). Annual incidence rates range from 0.2 to 0.8/1000, and mortality ranges from 17 to 48% within 30 days and from 19 to 30% within 6 to 12 months after an episode of acute PE. All this justifies the need to organize and conduct research aimed at improving early diagnosis and predicting the outcomes of the disease, as well as reducing morbidity and mortality from this pathology, justified from a scientific point of view and in demand from a practical point of view.

As a result of this study, based on the analysis of data from the Regional PE Registry (REPER) and the international register of hospitalized patients diagnosed with acute PE using multi-detector computed tomographic lung angiography (MDCT-PA), the authors examined the predictive value of the total number of leukocytes and hemoglobin levels at admission of patients with acute pulmonary embolism (PE) based on the European Society of Cardiology (ESC) mortality risk model.

Strengths of the study: sufficient sample of patients; well-planned study design; the use of adequate statistical methods, the availability of introducing the definition of the studied parameters and, in this regard, the high potential for practical use of the results, the presence of open questions that reasonably require further research in order to improve the risk stratification of adverse cardiovascular events associated with acute PE, for example, to clarify the role molecular biomarkers.

Weaknesses of the presentation of the manuscript: the lack of p-value of the parameters in Table 1. Moreover, the authors should add the data on the baseline levels of echocardiography and their relationship with the studied parameters.

Thank you very much for the review! We added the p values in table 1 as suggested.

We also added correlation between total leukocyte number and Hb concentration with some basic echocardiography parameters in table 3 and the appropriate comment in the results section.

Reviewer 2 Report

First, I would like to congratulate the authors on finishing this manuscript. It's an intriguing observation that has been observed in other studies thus far as well. My comments are below divided by sections.

1.      INTRODUCTION

a.      You should add more regarding the impact of leukocytosis on RV dysfunction. Autopsy studies have shown that an influx of inflammatory cells may contribute directly to RV dysfunction. (PMID: 18808531; 16814320; 12850410; 17646195; 23674436)

b.      Similarly, in the introduction, please add why anemia/low Hgb is essential to be considered a prognostic indicator in PE. Has any direct relationship been discovered so far? Any pathophysiological link so far to hypothesize higher adverse outcomes, specifically in PE patients, in addition to higher baseline risk of cardiac and renal dysfunction in chronic anemia patients.

2.      IN MATERIAL AND METHODS

a.      Were there any adjustments made regarding the use of thrombolytic therapy and anticoagulation therapy since this can have effects both on hgb and leukocytosis/inflammation (Heparin also has anti-inflammatory properties)

b.      Please consider adding Kaplan-Meier estimates for mortality concerning WBC and Hgb levels.

c.       How were the patients selected? Were ICD-9 or ICD-10 diagnosis codes used? Age cut-offs? Any exclusion criteria? Include a flow chart for patient selection.

3.      DISCUSSION

a.      The section needs to be expanded with more relevant references. Example PMID:23674436; 34846193

b.      More discussion should be included in critiquing results compared to previous studies on this topic.

4.      LIMITATION

a.      Depending on how patients/diagnosis was selected, please include selection bias. This study is also not generalizable to other parts of the world and should be included as a limitation.

b.      Another limitation of the study would be the lack of hypercoagulable workup (this is not an uncommon practice but should be mentioned as it could have an impact on WBC levels)

c.       Was the WBC and Hgb changes transient or persistent throughout the hospitalization or till discharge? Likely, this is not included in the study and should be part of the limitation.

d.      Any concomitant immunomodulator/anti-inflammatory/anti-cancer therapy in the patients would also impact WBC and Hgb levels. If data is unavailable, then this is also a study limitation and should be mentioned.

5.      CONCLUSION

a.      Also, leave a future direction statement. For example, the use of WBC and Hgb for PE risk stratification and the PESI should be tested in prospective studies before clinical adoption.

Author Response

First, I would like to congratulate the authors on finishing this manuscript. It's an intriguing observation that has been observed in other studies thus far as well. My comments are below divided by sections.

  1. INTRODUCTION
  2. You should add more regarding the impact of leukocytosis on RV dysfunction. Autopsy studies have shown that an influx of inflammatory cells may contribute directly to RV dysfunction. (PMID: 18808531; 16814320; 12850410; 17646195; 23674436)

Dear reviewer, thank so much for your valuable comments! We add the paragraph about the possible direct role of inflammation and leukocytes in RV dysfunction in patients with acute PE.

  1. Similarly, in the introduction, please add why anemia/low Hgb is essential to be considered a prognostic indicator in PE. Has any direct relationship been discovered so far? Any pathophysiological link so far to hypothesize higher adverse outcomes, specifically in PE patients, in addition to higher baseline risk of cardiac and renal dysfunction in chronic anemia patients.

Also very important point. We added also paragraph considering this possible relationship between anemia and outcome of acute PE patients.

  1. IN MATERIAL AND METHODS
  2. Were there any adjustments made regarding the use of thrombolytic therapy and anticoagulation therapy since this can have effects both on hgb and leukocytosis/inflammation (Heparin also has anti-inflammatory propertie

Patients treated with unfractionated heparin or low-molecular weight heparins, and a very small number of patients received fondaparinux. Thus, almost all patients were under heparins, and we have nothing to compare with. Direct oral anticoagulants were introduced 5-7 days later. Patients with renal failure were treated more often with unfractionated heparin and it would be biased to compare those patients with LMWHs.

On the other hand, we, added the prognostic value for the prediction hospital mortality separately in patients who were treated and not treated with thrombolysis, and it is highlighted in the separate section in the results. 

  1. Please consider adding Kaplan-Meier estimates for mortality concerning WBC and Hgb levels.

Thank you for this comment, we added KM estimates using quartiles of total leukocyte counts and Hb levels.

  1. How were the patients selected? Were ICD-9 or ICD-10 diagnosis codes used? Age cut-offs? Any exclusion criteria? Include a flow chart for patient selection.

Our registry enrolled all patients admitted to hospitals (in the majority of cases the admission was in the intensive care units) involved who had symptoms of acute PE confirmed at MDCT-PA. Exclusion criteria were very small sub-segmental PE, and patients who were admitted to hospitals for the treatment of their terminal diseases. We can say that this registry is the registry of consecutive PE patients and all of them had complete blood counts determined at the admission and the data about the hospital outcome. Because of that we don’t have to show anything interesting on the flowchart, but we described these inclusion and exclusion criteria more precisely in the methodology.  

We used ICD-10 codes classification, and there is only lower age limit – 18 years, it is added to the methodology.

  1. DISCUSSION
  2. The section needs to be expanded with more relevant references. Example PMID:23674436; 34846193
  3. More discussion should be included in critiquing results compared to previous studies on this topic.
  4. LIMITATION
  5. Depending on how patients/diagnosis was selected, please include selection bias. This study is also not generalizable to other parts of the world and should be included as a limitation.

We added this at limitation section.

  1. Another limitation of the study would be the lack of hypercoagulable workup (this is not an uncommon practice but should be mentioned as it could have an impact on WBC levels).

If I understood, you well, I add the comment of this in the limitation section too.

  1. Was the WBC and Hgb changes transient or persistent throughout the hospitalization or till discharge? Likely, this is not included in the study and should be part of the limitation.

In this investigation we did not use repeated findings of TLC and Hb and thus we do not investigate the role of the dynamic changes of CBC to acute PE outcome.

  1. Any concomitant immunomodulator/anti-inflammatory/anti-cancer therapy in the patients would also impact WBC and Hgb levels. If data is unavailable, then this is also a study limitation and should be mentioned.

Yes, thank you, we have data about pre-PE chemotherapy and radiation therapy but we did not want to exclude these patients, and we mention exact figures into limitation section about this matter.

  1. CONCLUSION
  2. Also, leave a future direction statement. For example, the use of WBC and Hgb for PE risk stratification and the PESI should be tested in prospective studies before clinical adoption.

We added recommended future directions into the end of the conclusion section.